

# Numerical ability and improvement through interindividual cooperation varied between two cyprinid fish species, qingbo and crucian carp

Wei Xiong*, Lian-Chun Yi*, Zhong-Hua Tang and Shi-Jian Fu

Laboratory of Evolutionary Physiology and Behavior, Chongqing Key Laboratory of Animal Biology, Chongqing Normal University, Chongqing, China
* These authors contributed equally to this work.

## ABSTRACT

We used qingbo (*Spinibarbus sinensis*) and Chinese crucian carp (*Carassius auratus*) to test whether numerical discrimination could be improved by the coexistence and possible cooperation of conspecies or heterospecies. We conducted a spontaneous shoal choice test of singletons, conspecific dyads and heterospecific dyads under different numerical comparisons (8 vs. 12, 9 vs. 12 and 10 vs. 12). Singletons of qingbo could discriminate only 8 vs. 12, whereas the dyads of qingbo showed better numerical acuity, as they could discriminate 10 vs. 12. Crucian carp may have poor numerical ability, as both singleton and dyads showed no significant preference for larger stimulus shoals, even at the 'easier' numerical discrimination, that is, 8 vs. 12. Furthermore, heterospecific dyads of crucian carp and qingbo did not show significant preference for larger shoals at any numerical comparison in the present study. It is suggested that both the numerical ability and the possibility for improvement by interindividual interaction and hence cooperation might vary among fish species, and the interaction between heterospecies in the present study showed negative effect on numerical ability possibly due to the different behavioural and cognitive traits which make the information transfer and consensus difficult to reach.

## INTRODUCTION

Many studies have demonstrated that the ability to discriminate between quantities is widespread (*Vallortigara, 2012, 2017*) and has been exhibited in preverbal infants (*Lipton & Spelke, 2003*) as well as in a range of nonhuman animals, such as dolphins (*Kilian et al., 2003*), dogs (*Ward & Smuts, 2007*), chickens (*Rugani, Regolin & Vallortigara, 2007, 2008*), tortoises (*Gazzola, Vallortigara & Pellitteri-Rosa, 2018*), frogs (*Stancher, Rugani & Vallortigara, 2015*), fish (*Agrillo & Dadda, 2007*; *Agrillo, Dadda & Bisazza, 2007*; *Stancher et al., 2013*; *Petrazzini et al., 2015*) and invertebrates (*Gross et al., 2009*; *Reznikova & Ryabko, 2011*). Numerical ability can improve antipredation efforts of

Corresponding author
Shi-Jian Fu, shijianfu9@cqnu.edu.cn

animals by allowing them to compare shoals and join larger ones because larger shoals confer the ecological benefits of improved survivorship under the risk of predation (*Foster & Treherne, 1981*; *Landeau & Terborgh, 1986*). Thus, it has been assumed that the ability to discriminate quantities and stay with a larger shoal is a vital advantage in fish species that live in groups.

Improved cognitive ability through interindividual cooperation was suggested more than one century ago (*Galton, 1907*), and the so-called 'collective intelligence' has frequently been proven both in the field and in laboratory studies (*Krause, Ruxton & Krause, 2010*; *Bisazza et al., 2014*). The improvement in numerical ability has also been found in grass carp (*Ctenopharyngodon idellus*) (*Bai, Tang & Fu, 2019*) and guppy (*Poecilia reticulata*) (*Bisazza et al., 2014*). The possible mechanisms might be due to the so-called 'meritocratic leadership', that is, the performance of a dyad is determined by the better member, which plays a leadership role in decision making (*Bisazza et al., 2014*). In nature, mixed-species shoals are frequently found in a wide range of animal taxa and provide advantages in terms of enhanced foraging efficiency and predator avoidance (*Goodale et al., 2010*; *Kleinhappel et al., 2016*). Recently, using two cyprinid fish species as experimental animals, we found that numerical ability might facilitate by cooperation between mixed-species individuals of grass carp (*Ctenopharyngodon idellus*) (good numerical ability) and Chinese bream (*Parabramis pekinensis*) (poor numerical ability) as the latter exhibited shutter behaviour between two stimulus shoals following those of grass carp and hence improved numerical ability compared to either singleton Chinese bream and its conspecific dyad (*Bai, Tang & Fu, 2019*). Furthermore, the preference for either larger or smaller shoals might vary between singletons and dyads due to some emotional and (or) motivational factor (*Regolin, Vallortigara & Zanforlin, 1995*). Thus, the aim of the present study was to test if the abovementioned improvement also manifests in other cyprinid fish species.

In the present study, we selected qingbo (*Spinibarbus sinensis*) and crucian carp (*Carassius auratus*) as the experimental animals. Both species prefer group living and are widely distributed throughout China. The numerical ability of qingbo (in singletons is 2:3) was recently documented (*Xiong et al., 2018*), while data on the numerical ability of crucian carp are unavailable. To fulfil our goal, we conducted a spontaneous shoal choice test of singletons and dyads (both conspecific and heterospecific) under a numerical comparison range of 8 vs. 12 to 10 vs. 12 (8 vs. 12 was used because almost all fish species showed a similar or higher numerical ability at a 2:3 ratio). We tested our hypothesis by comparing the preference to choose the larger stimulus shoal among singletons, conspecific and heterospecific dyads.

## MATERIALS AND METHODS

### Ethics statement

This study was approved by the Animal Care and Use Committee of the Key Laboratory of Animal Biology of Chongqing (Permit Number: Zhao-20161012-01) and was performed in strict accordance with the recommendations in the Guide for the Care and Use of Animal at the Key Laboratory of Animal Biology of Chongqing, China.

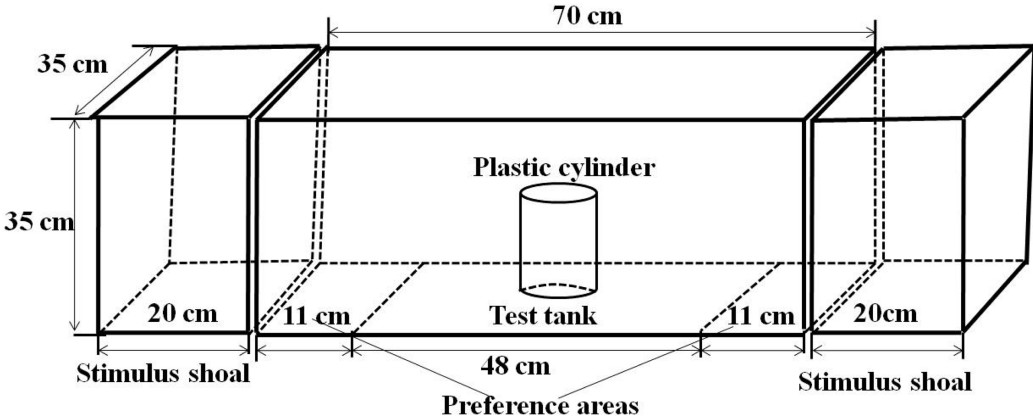

**Figure 1 Experimental setup showing the structure of experimental arena.**

## Subjects and housing conditions

Experimental juvenile qingbo ($N = 400$, standard body length $5.10 \pm 0.05$ cm; body mass $2.87 \pm 0.09$ g) and juvenile crucian carp ($N = 400$, standard body length $3.88 \pm 0.04$ cm; body mass $1.74 \pm 0.07$ g) were purchased from a local market. Prior to the experiment, both qingbo and crucian carp were randomly divided into two groups and reared separately as either the stimulus shoal or the test fish. All fish were reared in water tanks that included a biological filter with recirculating water for 4 weeks. During this period, the temperature of the dechlorinated water was maintained at $25 \pm 1$ °C, and the oxygen content was maintained above 90% saturation. The photoperiod cycle was 12 h of light:12 h of darkness. All fish were fed once daily at 8:00 am with a commercial floating diet (Tongwei Company, Chengdu, China). The uneaten food and feces were cleaned up using a siphon. Furthermore, the experimental measurements were performed between 8:00 and 18:00 after 4 weeks of acclimation.

## Experimental design

We conducted 360 spontaneous choice tests, that is, 20 repetitions of choice tests for singletons of qingbo, singletons of crucian carp, dyads of qingbo, dyads of crucian carp and dyads of mixed-species pairs (one qingbo and one crucian carp). Either a single (singleton test) or a pair of fish (dyad test) was used as the test fish for each choice test. The numerical comparison was 8 vs. 12, 9 vs. 12 and 10 vs. 12.

## Protocol and setup

The experimental apparatus used to assess the spontaneous shoal choice test consisted of a test tank ($70 \times 35 \times 35$ cm) with two stimulus tanks ($20 \times 35 \times 35$ cm) positioned at both ends (Fig. 1). Two 16-W fluorescent lamps were located at both ends of the stimulus tank. The tanks were filled with 10 cm of water. To prevent other interference factors, all of the walls of the test tank, except for the wall connected with the stimuli tank, were covered with white paper, and an opaque curtain was stretched around the arena. During the test, the behavioural responses of the test fish were recorded using a webcam
(Logitech Pro 9000; Logitech Company, Suzhou, China) located 100 cm over the test tank and above the arena and connected to a remote computer.

To avoid foraging behaviour, fish were fed to apparent satiation 2 h before the measurements. Stimulus fish were first introduced to the stimulus tank 5 min prior to the test. The test fish were then introduced to the centre of the test tank with a holding device (plastic cylinder with 10 cm of diameter and 15 cm of height) and remained in the device for 3 min. After that period, the holding device was gently removed, and the position of the test fish was recorded for 15 min at 15 frames/s. Each individual test fish was used only once, whereas individuals of the stimulus shoals may have been used more than once. A total of 20 repetitions of each treatment were recorded. In the numerical preference assessmentfor singletons or conspecific dyads, all stimulus shoal members were the same fish species. In the numerical preference assessmentfor heterospecific dyads, half qingbo and half Chinese crucian carp formed a mix stimulus shoals of two species for 8 vs. 12 and 10 vs. 12 numerical comparison, whereas 10 groups of five qingbo mixed with four Chinese crucian carp and 10 groups of four qingbo mixed with five Chinese crucian carp for stimulus shoal of nine individuals. The positions of the test fish were analysed by the automated tracking software program idTracker (v 2.1) (*Pérez-Escudero et al., 2014*; *Tang et al., 2017*). In conspecific dyads, the video files of two of the measurements of 8 vs. 12 in qingbo, three of the measurements of 9 vs. 12 in crucian carp and one measurement of 10 vs. 12 in crucian carp failed to be decoded. Therefore, the final repetitions were 18, 17 and 19 in 8 vs. 12, 9 vs. 12 and 10 vs. 12, respectively.

The preferences of the test fish were defined as the time spent by the test fish within the 11 cm preference zones during the 15-min periods of recording (i.e. within 11 cm of the wall adjacent to the stimulus tanks on either side) and were calculated as follows: observations within the preference zone of either the larger or smaller shoal/(observations within the preference zone of the larger shoal + observations within the preference zone of the smaller shoal). In conspecific dyads, we measured each fish in the dyad independently and then calculated the average for dyads.

### Statistical analysis

Statistical tests were carried out using SPSS 11.5 (SPSS, Armonk, NY, USA). The time spent in the preference zone was recorded as a measure of the preference of each test fish for either the larger or smaller stimulus shoal. All data were tested for normality by a one-sample Kolmogorov–Smirnov test. The linearly mixed model (tested number as random factor) was used to test the significance of the preference for larger or smaller shoals. All values are presented as the mean ± S.E., and $P < 0.05$ was used as the level of statistical significance.

## RESULTS

### Experiment 1 numerical ability of singletons

Qingbo showed a significant preference for larger shoals at only a numerical contrast of 8 vs. 12 ($F_{1,38} = 6.083$, $P = 0.018$), that is, qingbo were able to discriminate at only a 2:3 ratio (Fig. 2). No significant preference was observed for 9 vs. 12 ($F_{1,38} = 0.119$, $P = 0.732$), or 10 vs. 12 ($F_{1,38} = 1.742$, $P = 0.195$). Crucian carp showed no significant preference for

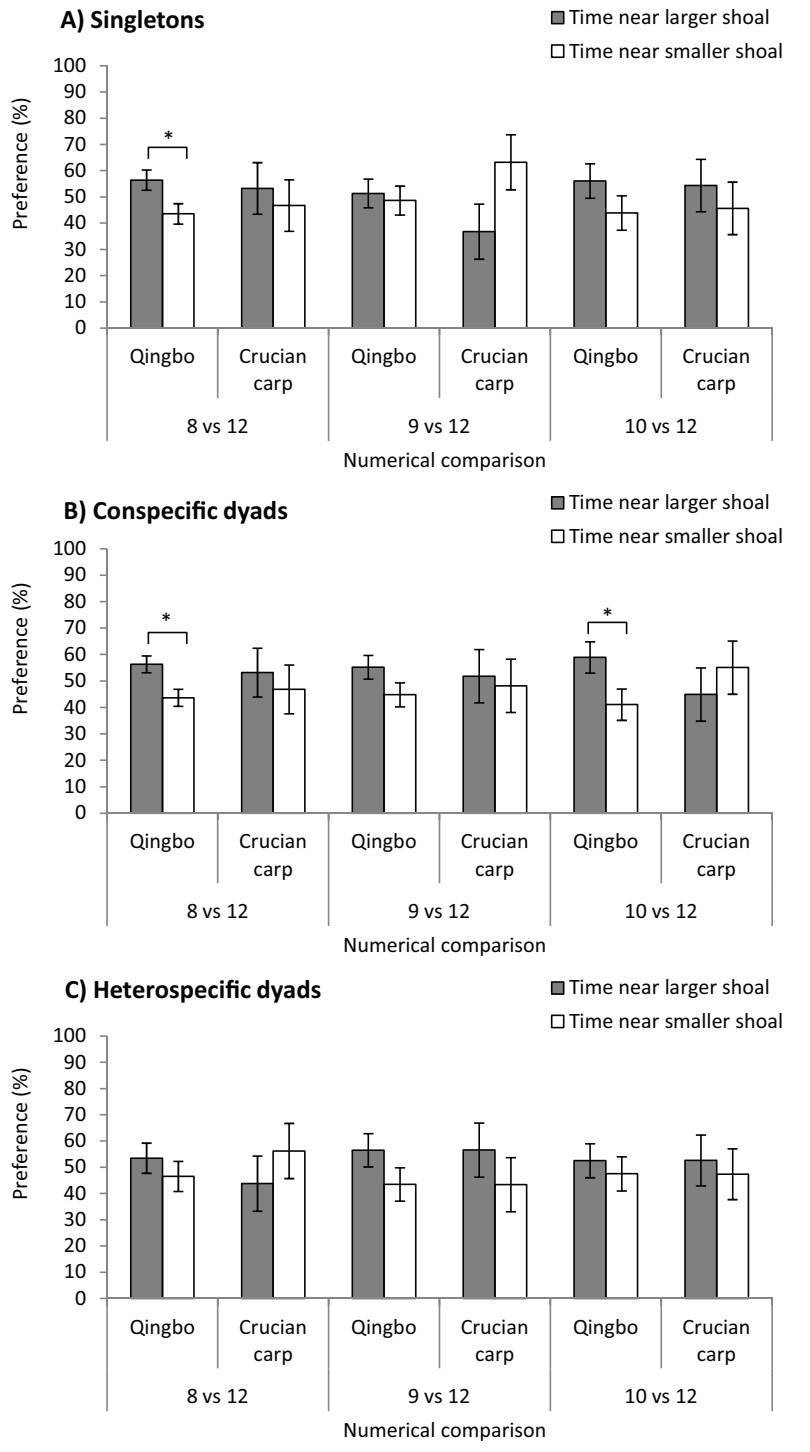

**Figure 2 The shoal test of qingbo and crucian carp in both singletons (A) and dyads (both conspecific and heterospecific) (B and C) under different numerical comparisons.** Mean ± S.E., 20 repetitions, except $N$ = 18 for 8 vs. 12 of conspecific dyad of qingbo, 17 for 9 vs. 12 and 19 for 10 vs. 12 of conspecific dyads of crucian carp. *Indicates significant preference for larger shoals ($P$ < 0.05).

larger shoals at any numerical comparison: 8 vs. 12 ($F_{1,38} = 0.223$, $P = 0.640$), 9 vs. 12 ($F_{1,38} = 3.741$, $P = 0.061$) or 10 vs. 12 ($F_{1,38} = 0.335$, $P = 0.567$).

## Experiment 2 numerical ability of conspecific dyads

Dyads of qingbo showed a significant preference for the larger shoal at numerical contrasts of 8 vs. 12 ($F_{1,34} = 7.874$, $P = 0.008$) and 10 vs. 12 ($F_{1,38} = 4.581$, $P = 0.039$) but not at 9 vs. 12 ($F_{1,38} = 2.662$, $P = 0.111$). However, no significant preference was observed at any numerical comparison in crucian carp: 8 vs. 12 ($F_{1,38} = 0.235$, $P = 0.631$), 9 vs. 12 ($F_{1,32} = 0.064$, $P = 0.801$) or 10 vs. 12 ($F_{1,36} = 0.511$, $P = 0.479$).

## Experiment 3 numerical ability of heterospecific dyads

Neither qingbo nor crucian carp showed any significant preference for larger shoals at any numerical comparison in the preset study; 8 vs. 12 (qingbo: $F_{1,38} = 0.818$, $P = 0.371$, crucian carp: $F_{1,38} = 0.704$, $P = 0.407$), 9 vs. 12 ($F_{1,38} = 2.118$, $P = 0.154$, $F_{1,38} = 0.969$, $P = 0.331$) and 10 vs. 12 ($F_{1,38} = 0.297$, $P = 0.589$, $F_{1,38} = 0.155$, $P = 0.696$).

## DISCUSSION

The main aims of the present study were to test whether interaction (i.e. cooperation) between individuals could facilitate the numerical abilities of two cyprinid fish species and whether the interaction between different fish species could achieve similar performance compared to the possible improvement between conspecific dyads. The numerical ability indicated by shoal preference increased from a threshold of 8 vs. 12 in singleton qingbo to at least 10 vs. 12 (the most subtle numerical difference in this study) in qingbo dyads. However, crucian carp may show poor numerical ability as it showed no preference to larger shoal in any numerical comparison in the present study. Thus, the possible improvement in numerical ability between individual crucian carp and heterospecific dyads was difficult to assess; however, the coexistence of heterospecies did show a negative effect on the numerical ability of qingbo, as the qingbo in heterospecific dyads could not even discriminate in the numerical comparison of 8 vs. 12.

### Qingbo showed greater numerical ability indicated by shoal preference than crucian carp

Given the ecological benefit of collective living, half of all fish species prefer group living throughout life or for at least a part of their life history (*Shaw, 1978*). Thus, numerical discrimination may play an important role in fish species (*Petrazzini et al., 2015*), and such ability has been widely documented in fish species (*Agrillo, Dadda & Bisazza, 2007*; *Potrich et al., 2015*; *Xiong et al., 2018*). In the present study, singletons of qingbo could discriminate in the numerical comparison of 8 vs. 12, which fall within the data of other fish species (*Lucon-Xiccato & Bisazza, 2017*, which usually varied from 0.5 to 0.75). However, crucian carp showed no significant preference for larger shoals in any of the numerical comparisons in the present study. It is suggested that differences in numerical ability might exist in closely related fish species. This may be reasonable, as qingbo in the field prefer to live in fast-flowing water and exhibit a group-living lifestyle

(*Killen et al., 2016*; *Tang et al., 2017*). However, crucian carp prefer to live in still water with a more complicated habitat, and selection stress may be more severe on characteristics other than group living and numerical ability. However, cognitive and behavioural traits may also be involved in such differences in the artificial experimental conditions, as the crucian carp seemed much less bold and showed fewer exploration activities than qingbo. According to our observation (not directly recorded), it seemed that Chinese crucian carp performed considerably fewer shutters between two stimulus shoals than qingbo in the present study. Further investigation on boldness and stress hormone such as plasma cortisol level during choice test might provide useful information. Furthermore, the data in Fig. 2 clearly show that the mean preference ratio of crucian carp singletons, under the numerical comparison of 8 vs. 12 (possibly also under the 8 vs. 12 and 9 vs. 12 conspecific dyads), was similar to that of the qingbo. It is suggested that the large interindividual variation might mask the possible shoal preference of crucian carp under these situations. The reason might be that some individuals were rather shy and would stay with the smaller stimulus shoal once they made the wrong choice (*Lucon-Xiccato et al., 2017*). Therefore, we cannot tell if they cannot discriminate in the numerical comparison, if the test fish dare not make a choice or if they do not care about the difference under the conditions of the present study. Perhaps a further study under a food choice test might provide more useful information, as a previous study found that the numerical ability varied between different ecological contexts (*Lucon-Xiccato & Dadda, 2017*). Nevertheless, the present study suggested that the numerical discrimination threshold of singleton qingbo is approximately 2:3, whereas the threshold of crucian carp is less than that if it has any numerical ability or shoal preference.

## The dyads of qingbo showed better numerical acuity than sigletons

Increased cognitive ability through interindividual interaction was again demonstrated in the present study, as the numerical threshold of qingbo dyads improved profoundly when compared to that of singletons. The previous studies in grass carp (*Bai, Tang & Fu, 2019*) and guppy (*Bisazza et al., 2014*) found both species can discriminate at a ratio of 0.75 (i.e. 3:4) by interindividual cooperation. In the present study, dyads of qingbo could even discriminate in the 10 vs. 12 comparison (i.e. 0.833). Given that it is the smallest numerical comparison treatment designed in the present study, it is suggested that qingbo may achieve a considerably high numerical discrimination performance through cooperation. However, it was worth noting that dyads of qingbo fail to discriminate 9 vs. 12, even though the preference for larger stimulus shoals was 23% higher than that for smaller shoals. It is possible that the aforementioned large interindividual variation might mask the significant difference. The improvement in numerical ability by interindividual cooperation has also been found in grass carp (*Ctenopharyngodon idellus*) (*Bai, Tang & Fu, 2019*). Given the profound improvement in numerical ability, it is possible that the 'meritocratic leadership' might be involved in cooperation during the numerical discrimination process. However, it has long been suggested that the emotional and (or) motivational factor related to the numerical discrimination and spatial

cognition might also be involved in the improved performance between dyads and singletons (*Regolin, Vallortigara & Zanforlin, 1995*).

Both singletons and dyads of crucian carp showed no shoal preference at any shoal preference in the present study; thus, we cannot draw any conclusions about the possibility for improvement in numerical ability by the interindividual interaction of crucian carp. However, because qingbo could improve from the numerical comparison of 8 vs. 12 in singleton to 10 vs. 12 in dyad, if such improvement does exist in crucian carp, the threshold of singleton crucian carp must be considerably lower than 7 vs. 12 (which is the best possible numerical contrast threshold of the crucian carp dyad). Again, the spontaneous shoal choice test under another ecological context (e.g. mating or foraging) or a training protocol used before might provide more useful information (*Lucon-Xiccato & Bisazza, 2017*).

## Qingbo in heterospecific dyads showed poor numerical ability even lower than singletons

In the present study, the crucian carp in the heterospecific dyads showed no shoal preference in any numerical comparisons. However, because we did not conduct an easier numerical comparison than 2:3, we cannot determine the thresholds of singleton or dyads of crucian carp, if they have any. Thus, we also cannot draw any conclusions about the possibility of the numerical ability (or shoal preference) of crucian carp increasing through heterospecific cooperation. However, on the contrary, the company of crucian carp showed a negative effect on the numerical ability of qingbo, as qingbo in heterospecific dyads could not even discriminate in the 8 vs. 12 comparisons (the threshold of singletons). These mechanisms need further investigation. Because these two fish prefer living in different habitats, the behavioural and cognitive traits might be completely different, thus making the information transfer and consensus difficult to reach, which may be the precondition of the collective intelligence. Further investigation using fish species of naturally mixed shoals as experimental models might give more useful information.

## CONCLUSIONS

Qingbo showed a similar numerical acuity to previous studies with a threshold of 0.66, which could be facilitated to at least 0.83 by interindividual cooperation. Crucian carp showed no shoal preference in the present study, possibly due to low cognitive ability, large interindividual variation in numerical ability, low boldness or no motivation for shoal preference, or improper numerical comparison (i.e. only large numerical comparison), which might make this a poor experimental model for numerical ability research. The interaction between heterospecific dyads elicited a decreased numerical ability of qingbo, which became even lower than the threshold of singletons. However, it is difficult to draw any safe conclusions about the difference in cooperation efficiency on numerical comparisons between conspecific and heterospecific dyads due to the poor performance of crucian carp and/or the lack of small numerical comparisons in the present study.

## ACKNOWLEDGEMENTS

We would like to thank Dr. Jennifer Vonk and two anonymous reviewers for insightful comments and constructive suggestions that greatly improved the manuscript.

### Funding

This work was supported by the National Natural Science Foundation of China (No. 31670418). The funders had no role in study design, data collection and analysis, decision to publish, or preparation of the manuscript.

### Grant Disclosure

The following grant information was disclosed by the authors:
National Natural Science Foundation of China: 31670418.

### Competing Interests

The authors declare that they have no competing interests.

### Author Contributions

- Wei Xiong conceived and designed the experiments, performed the experiments, analysed the data, prepared figures and/or tables, authored or reviewed drafts of the paper.
- Lian-Chun Yi conceived and designed the experiments, performed the experiments, analysed the data, prepared figures and/or tables, authored or reviewed drafts of the paper.
- Zhong-Hua Tang performed the experiments, analysed the data, prepared figures and/or tables.
- Shi-Jian Fu conceived and designed the experiments, performed the experiments, analysed the data, contributed reagents/materials/analysis tools, prepared figures and/or tables, authored or reviewed drafts of the paper, approved the final draft.

### Animal Ethics

The following information was supplied relating to ethical approvals (i.e. approving body and any reference numbers):

Animal Care and Use Committee of the Key Laboratory of Animal Biology of Chongqing (Permit Number: Zhao-20161012-01).

### Data Availability

The raw measurements are available in File S1.

### Supplemental Information

Supplemental information for this article can be found online at http://dx.doi.org/10.7717/peerj.6619#supplemental-information.

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
