# Peer review of "Numerical ability and improvement through interindividual cooperation varied between two cyprinid fish species, qingbo and crucian carp"

_PeerJ, doi:10.7717/peerj.6619_

## Round 0.1 · original submission · Minor Revisions

I received two expert reviews of your MS. One of the reviews was quite positive with the reviewer suggesting a number of points of clarification and minor corrections. The other reviewer was more critical and has several amendments and corrections to your literature review. Both raise possible issues of pseudoreplication, and ask for clarification regarding your statistical approach. You mention that each fish was tested only once, so I think you might just expand on how fish were identified and eliminated from multiple tests. However, both reviewers also find merit in your work and thus, I would like to invite you to submit a revision addressing their concerns. I agree that the study is of interest and will make a contribution to the literature but I have a few comments of my own:

In the abstract, strictly speaking, you didn’t “conduct a… choice..” You may have conducted a test in which the fish made a choice. Please reword.

Is it accurate to say that they could not discriminate because they did not show a preference? These are two different things. If there was no benefit to discriminate, you cannot conclude that a lack of preference reflects an inability to discriminate quantities. You address this is in the discussion, but be careful then how you word this in the abstract.
What do you mean, “showed poor cooperation on numerical ability.” This doesn’t make sense as written. I don’t think the construct of cooperation is well defined here. It isn’t clear what the rationale is behind the manipulation of presenting singletons and dyads. Please expand in the introduction. In general, the introduction is very lean. It would be helpful to expand more on the species differences and expectations, and the background for the methodology used here. Is it really that fish “cooperate” to perform better, or is it just that their preferences differ when alone or not alone? Could you please say more about what it means for quantity discrimination to be “improved by cooperation” (line 48)?
Lines 53-54, I think an “if” is missing between “test” and “the abovementioned…” manifests should be plural.
Please be sure to call out your Figures within the text.
On line 154, Agrillo is missing an o. Please check spelling carefully throughout.
Replace “since” with “because” when using in a non temporal context.

Reviewer 1 ·

Basic reporting

Language could be improved.

Experimental design

Statistics and data use should be explained and checked.

Validity of the findings

OK

Additional comments

The study reported in the MS “The numerical ability and improvement through interindividual cooperation varied between two cyprinid fish species, qingbo and crucian carp” is interesting and well-performed. I think this study will be of interest for the readers of PeerJ.
Below I provided a list of suggested changes and pints that could be improved to better present the study.

L19-21
We used qingbo (Spinibarbus sinensis) and Chinese crucian carp (Carassius auratus) to test whether numerical discrimination could be improved by the cooperation of conspecific or heterospecific dyads of fish.

L24
Numerical cognition should be ”performance” or “numerical acuity”

L25
Remove “under the condition”

L27
....even at the ‘easier’ numerical discrimination, i.e., .....

L39
Remove “other”

L42-45
In some cases being in the larger group increase competition for food and it is not beneficial. I think it is better to rewrite this sentence considering only the antipredator defence advantage.

L47
Prefer should be “shows”

L47
Break the sentence after “living”. Then start a new sentence for the cooperation.

L53
Which species have been tested by Bai?

L62
“We tested” and “the ability to choose the larger stimulus”

L85
How many repetitions overall?

L94-96
During the test, the behavioral responses of the test fish were recorded using a webcam (Logitech Pro 9000; Logitech Company, Suzhou, China) located 100 cm over the test tank and above the arena and connected to a remote computer.

L99
Explain the holding device

L113
IMPORTANT POINT: I don’t understand how the authors dealt with the data of the dyads. Did they measure each fish in the dyad independently and then they calculated the average? Please clarify in detail this important information.

L117-118
It seems that the authors used an F test (repeated measures ANOVA?) rather than a t test. Please check carefully the analysis.

L154
Agrillo

L163
Psychological should be cognitive

L163-165
Can the authors provide some data for the interesting boldness effect? Maybe they can also run a very brief experiment to measure boldness....

L169-171
I firstly read about this idea in the paper:
Lucon-Xiccato, T., Dadda, M., Gatto, E., & Bisazza, A. (2017). Development and testing of a rapid method for measuring shoal size discrimination. Animal Cognition, 20, 149-157.
Maybe the authors can cite it here.

L182
Bisazza
Please double check references

L216
Psychological should be cognitive

L273
Danio rerio in italic

Figure 2
I think the asterisk should be on the top of the two bars because the statistical test was for the difference between the two bars.

Reviewer 2 ·

Basic reporting

The English Level is good throughout the manuscript. The structure of the article is clear and follow a precise workflow.
The Raw data file provided is clear, but it would be helpful to add some labels that better explain what each data group refers to.
The Figures are simple and clear.


- Line 41: please add birds and tortoises to the list, e.g.
https://royalsocietypublishing.org/doi/10.1098/rsbl.2018.0649
https://www.ncbi.nlm.nih.gov/pubmed/18665721
https://www.ncbi.nlm.nih.gov/pmc/articles/PMC3171108/
https://www.ncbi.nlm.nih.gov/pubmed/17227192
https://journals.plos.org/plosone/article?id=10.1371/journal.pone.0065262
https://royalsocietypublishing.org/doi/full/10.1098/rspb.2009.0044

- Line 42: Given that the comparative literature on this topic is large and examples cannot cover all the field, it would be important to provide the readers with some general reviews, I would recommend e.g. Vallortigara, G. (2017). An animal’s sense of number. In “The nature and Development of Mathematics. Cross Disciplinary Perspective on Cognition, Learning and Culture” (Adams, J.W., Barmby P., Mesoudi, A., eds.), pp. 43-65, Routledge, New York.
https://www.tandfonline.com/doi/abs/10.1080/02643294.2012.654772
https://www.ncbi.nlm.nih.gov/pmc/articles/PMC5784035/
https://royalsocietypublishing.org/doi/pdf/10.1098/rstb.2017.0120

Some of the reference reported are not appropriated for the sentence to which they refer. For example:
- Line 45: the reference Miletto Petrazzini and colleague (2015) is not correct if referred to shoals comparison and the discrimination of the larger one. The cited study investigated ordinal numerical abilities using a learning procedure with food as a reward. Cite this study in the list of numerical evidence at line 41
- Line 183: The study of Potrich, Sovrano, Stancher & Vallortigara (2015) does not provide evidence about interindividual cooperation and “collective intelligence”. I suggest to move this citation in the evidence of numerical discrimination of groups of conspecifics.

Experimental design

It is unclear to me how data for dyads were analyzed. It seems that individual data for each dyad were used, thus raising a serious problem of pseudo-replication. If not, the authors should explain how data for dyads were computed (one animal selected at random in each dyad?).

I suggest to provide more information about the set up and the procedure used for each condition. For example, in the heterospecific dyads, were the two shoals formed by a mix of the two species (half of qingbo and half crucian carp)?

Validity of the findings

There is a contradiction in the discrimination results of singletons of qingbo fish. In the graph and the manuscript the authors indicated that singleton of qingbo discriminate all the three comparison proposed (8 vs.12, 9 vs. 12 and 10 vs.12), but in the result section (line 130-131) and in the raw data the authors indicated that qingbo fish fail in the 9 vs.12. I suggest to correct these misunderstanding. Moreover, if it is correct that qingbo fish fail in the 9 vs.12, the authors should discuss about this.

It is unclear to me why the authors use the term “cooperation of conspecific” to account for changes in performance in numerical discrimination. A simple and more parsimonious hypothesis is that emotional/motivational variables can account for change in performance without involving anything ‘cognitive’ (see as an example in a different domain, that of object permanence and spatial cognition, e.g. https://www.sciencedirect.com/science/article/pii/S0003347285702321?via%3Dihub.

- Line 159: I would be more cautious in attributing or denying quantitative abilities to any particular species. The authors are testing here quantitative abilities during shoaling and it could be that carps are able to make quantitative discrimination under different testing conditions, i.e. using food or in general under associate learning procedures. Furthermore, the raw data show that a lot of carp subjects spend the entire time close to one group (irrespective of whether it is larger or smaller) without exploring the environment, suggesting that they don’t pay enough attention to the two shoals.

Additional comments

The Discussion is very long and repetitive; I would suggest shortening it.

- Line 55: To my knowledge, Crucian carp is not the species Carassius auratus as the authors indicated in the manuscript, but Carassius carassius.

- Line 58: please provide reference about previous studies on numerical ability in qingbo.

- Line 185-186. It is unclear to which species the authors are referring to in the sentence “However, both species can discriminate at a ratio of 0.75…”, since the results discussed in this paragraph were only about qingbo fish.

- Line 222: The authors demonstrated that singleton of qingbo fish have a threshold of 0.67 (since they discriminate only 8 vs. 12), not 0.75 as indicated.

Some Authors name reported throughout the manuscript are grammatically wrong or some authors are missing (e.g. Bisazza at line 182, Agrillo at line 154, Potrich, Stancher, Sovrano & Vallortigara al line 183). Please check and correct.

---

## Round 0.2 · Minor Revisions

Thank you for addressing the reviewers’ comments in your revision. I still have some issues I would like to see addressed before I can render a final decision on the MS.

On line 24, I would strike the word “profound.” I would also not use the term, “improvement” given that you did not test singletons first and then test the same fish in dyads. Throughout, you could change “improve” to “benefit from” or “facilitate”, such as on lines 58, 160, 204, 205.
Why is there a – on line 43?

Delete “the” on line 58.

On line 61, change “stumulus” to “stimulus” and insert a space after shoals.

I still think you should avoid talking about “ability” to choose (e.g., line 74) as if there is a correct/incorrect response when really it’s just a preference.

I still find the description of the procedure confusing. It is not clear what the shoals were made up of for each test fish condition (e.g., always mixed species or sometimes only conspecifics)? If this was a variable, why is it not accounted for in results? It seems to be a variable in Figure 2 but this is not described clearly in the text.

Why use only large quantities (8 or more) without assessing possible preferences for smaller numbers given the extensive literature on different systems for quantifying small versus large numbers? If group-living is preferable to living alone, perhaps the difference between 1 and 4 is more relevant than the difference between 8 and 12? You do address this briefly in the Discussion, but I think this consideration should temper some of your conclusions about the lack of preference of carp.

---

## Round 0.3 · Minor Revisions

I have reviewed your tracked manuscript and rebuttal letter and I would still like you to improve the clarity of your writing in the following places.
“Increase” is just as problematic as “Improvement” for the same reasons. Please change to “better” on line 24 of the tracked change version. On line 213, also change “increased” and “numerical ability.” See also line 259.

On line 61, use “be facilitated by” rather than “facilitate.”

Again, given that you are assessing a preference, not testing an ability, please strike reference to “numerical ability test” and refer to as “numerical preference assessment” or something along those lines. Please check carefully throughout.

You do not need to thank me for suggestions, as this is the role of editor and expected.

---

## Round 0.4 · accepted · Accept

Thank you for attending to these last final important edits. Please ensure proper spacing between words during the proofing stage as I noticed some words missing spaces between them in the tracked sections.

#